# Prediction Model for Compressive Strength of Porous Concrete with Low-Grade Recycled Aggregate

**DOI:** 10.3390/ma14143871

**Published:** 2021-07-11

**Authors:** Junshi Liu, Fumin Ren, Hongzhu Quan

**Affiliations:** 1School of Civil Engineering, Beijing Jiao Tong University, Beijing 100044, China; 13625429008zhang@163.com; 2School of Architectural Engineering, Qingdao Agricultural University, Qingdao 266109, China; quanhongzhu2006@qau.edu.cn

**Keywords:** recycled aggregate, construction waste, porous concrete, compressive strength, performance prediction model

## Abstract

As the first batch of products after the resource utilization of construction and demolition waste, low-grade recycled aggregate (RA) has not been fully utilized, which hinders the development of the comprehensive recycling industry of construction waste. Therefore, this paper studies the mechanical properties of porous concrete (POC) with low-grade RA. An improved relationship between porosity and compressive strength of brittle, porous materials is used to express the compressive strength of POC with recycled aggregate (RPOC), and the prediction for compressive strength of porous concrete with low-grade RA is constructed by analyzing the mechanism of compressive damage. The results show: the compressive strength of porous concrete decreases with the addition of low-grade recycled aggregate, but the effect is not obvious when the replacement rate is less than 25%. The error range of the relationship between porosity and compressive strength of RPOC is basically within 15% after improvement. The prediction model for compressive strength based on the ideal sphere model of aggregate can accurately reflect the compressive strength of porous concrete with low-grade RA. The results of this study can provide a reference for the staff to learn about the functional characteristics of recycled products in advance and provide security for the actual project.

## 1. Introduction

According to statistics, the average annual production of construction waste in the world has exceeded 8 billion tons [1], and China alone produces no less than 3.5 billion tons of construction waste every year, accounting for 30%~40% of the total urban waste [2]. This not only takes up a lot of land for landfill storage but also causes serious pollution of land, air, and water resources. In addition, with the rapid development of the construction industry, a large number of natural aggregates (NA) are mined, and the consumption is expected to reach 66 billion tons in 2025 [3]. In order to solve the problems of construction waste treatment and depletion of natural resources, the technology of recycled aggregate (RA) and recycled concrete has been widely studied and applied and has become a hot spot in various fields.

RA is obtained by crushing construction and demolition waste. It is similar to natural aggregate in function and use and is usually used in road traffic and the construction industry [4]. Because it can be recycled, researchers conducted a lot of research work and agreed that most of the RA surface is attached with old cement paste, which has the characteristics of high water absorption, large voids and high crushing value, so it is not suitable for high strength engineering [5,6]. In order to improve the recycling value of construction waste, the production and processing technology of high-grade RA was researched and developed. Two common production processes for RA are shown in Figure 1, in which the strengthening process represents mechanical grinding, polymer modification, mineral filling, etc. [7,8,9]. There is a big difference in performance between the recycled aggregate produced by winnowing and washing and the recycled aggregate obtained by simple crushing and screening, which is reflected in the apparent density, water absorption, porosity and other aspects [10,11,12,13]. According to these characteristics, the recycled aggregate is divided into different grades. The physical and mechanical properties of high-grade recycled aggregate are close to those of natural aggregate, so it is widely used in the preparation of building materials, bridge engineering and housing structure engineering [14,15]. However, the special production process not only improves the performance of recycled aggregate but also consumes energy and is easy to pollute the environment. For example, an additional 60% of the global warming potential and 61% of the acidification potential will be produced by using winnowing and washing processes [16], which is contrary to the original intention of protecting the environment and saving resources. On the other hand, the production process of low-grade RA is simple, but due to its weak performance, it is piled up and landfilled or used as the raw material of recycled products with a low replacement rate, which has become one of the construction waste by-products with a large stock at present [17]. Therefore, it is necessary to explore effective disposal ways to improve the utilization rate of low-grade RA.

Porous concrete (POC) is a new type of concrete material with a continuous porous structure, which is composed of coarse aggregate or a small amount of fine aggregate, cementing material and water. Its internal porosity can reach 20%~30%, with good air and water permeability, which can adapt to plant growth and increase the environmental benefits of the project [18,19,20]. However, the unique porous structure also limits the overall mechanical properties, causing this material to be mainly used in low-strength engineering. In recent years, POC is more and more popular in our life and is often used in some projects such as parking lots, sidewalks, river slope protection and roof greening [21,22]. POC with low-grade RA refers to POC made entirely or partially using low-grade RA. This measure can give full play to the environmental protection effect of POC and increase the utilization rate of construction waste, which will help to save natural resources and reduce environmental pollution.

Mechanical properties are important parameters that determine whether concrete materials can be applied to engineering. The functions of water permeability, water purification and sound absorption of POC should be established on the basis of meeting the strength requirements [23,24]. In order to reduce the construction cost and ensure the quality and safety of the project, the researchers explored the prediction model for compressive strength of POC based on the structural characteristics of the material. Chindaprasirt et al. found that the performance of POC is closely related to cement paste. The properties of cement paste are related to water-cement ratio, admixture and mixing time. It was also proved that the compressive strength of POC could be predicted by the strength equation of brittle, porous materials [25]. *Lian* et al. further explored the relationship between the permeability coefficient and the compressive strength of POC, and based on the characteristics of porous materials, analyzed the relationship between the compressive strength and porosity of POC with the characterization experience method, and proposed a compressive strength model derived from Griffith’s theory [26]. *Wang* et al. found that the thickness of the coating layer of the coarse aggregate determines the pore structure characteristics and the performance of POC to a certain extent. They also used three-dimensional and mathematical morphology-based methods to extract pore structure characteristics and obtained the relationship between the mechanical properties of POC and the thickness of the coating [27]. Zhong and Wille found that the compressive strength of porous concrete was affected by the strength of cement matrix, the dosage of admixture and the particle size of aggregate, and proposed a relationship between effective porosity and compressive strength based on the existing empirical prediction formula [28]. Zhang et al. used least squares support vector regression to simulate the highly nonlinear relationship between the performance and components of POC and obtained a more accurate compressive strength prediction model [29]. It can be seen that there are a variety of methods for predicting the compressive strength of POC, and the appropriate method must be selected according to the specific situation to ensure the accuracy of the results.

As we all know, compared with natural aggregates, there are weaker mechanical properties and greater attrition of recycled aggregates [30], which means that traditional material models are not necessarily suitable for recycled aggregate products [31]. At present, studies have shown that the mechanical properties of porous concrete with recycled aggregate (RPOC) are defective, but there are relatively few on strength prediction, and most of them are general mathematical relationships established by numerical fitting. Aliabdo et al. found that RA can increase the degradation of POC and established a general relationship between compressive strength and other parameters [32]. Hatanaka et al. studied the influence of the shape and strength of recycled aggregate on the compressive strength of POC and found that the strength and replacement rate of recycled aggregate are important factors [33]. Rasiah et al. found that the particle shape of RA has little effect on the compressive strength of POC and obtained the empirical relationship between compressive strength, porosity and permeability [34]. Lu et al. used a waste glass cullet to replace 50% of the aggregate to prepare POC and found that the addition of broken glass was not conducive to the structure and strength of concrete, but it could meet the engineering requirements when the mix proportion was reasonable [35]. Zhang et al. found that the crushing value of recycled aggregate was an important factor affecting the compressive strength, flexural strength and elastic modulus of POC, and when the crushing value was greater than 24%, the properties of POC decrease significantly [36]. Due to the difficulty of predicting the compressive strength of RPOC, some scholars even use the optimization decision technology of artificial intelligence to solve such problems. Naderpour and Mirrashid proposed an efficient prediction framework for compressive strength of POC based on a neuro-fuzzy algorithm and found that the compressive strength of concrete could be improved when the recycled aggregate was increased to 1000 kg/m^3^ [37]. Chen et al. established a prediction model of raw materials and compressive strength based on the back-propagation neural network method [3].

Previous studies are listed in Table 1. From the above literature, it can be seen that the complex types and replacement rate of aggregates make it difficult to form a universal prediction formula for the compressive strength of RPOC, which limits the development and promotion of products. Therefore, the RPOC needs to be reasonably classified and studied. There are two purposes of this study:To study the influence of low-grade recycled aggregate on the compressive strength of POC. The proposed relationship is incorporated into the compressive strength and porosity models of brittle, porous materials, and the validity of the method is verified through error analysis;Establish a simple and universal prediction model for compressive strength of POC with low-grade RA by analyzing the mechanism of compressive damage.

**Table 1 materials-14-03871-t001:** Summary of previous studies.

Authors	Type of Cementitious Material	Volume of RA (%)	Tests Performed	Main Findings
Chindaprasirt et al.	Normal Portland cement W/B of 0.20~0.36	0	Compaction and strength. Void distribution with height. State of bottom surface and strength.	*σ* = *σ*_0_exp (−*bV*)
Lian et al.	Ordinary Portland cement W/B of 0.30~0.38	0	Compressive strength. Porosity	* σ * = *B* $\sqrt{{\left(1-V\right)}^{m}e^{-np}}$
Wang et al.	Ordinary Portland cement W/B of 0.20	0	Specific surface area of coarse aggregate. Image processing and analysis to extract the pore structure.	The optimum thickness of cement paste of porous concrete is 590 μm.
Zhong and Wille	White cement W/C of 0.22~0.55	0	Compressive strength. Effective porosity.	* σ * = *σ*_0_(1 − *mφ*)($\frac{d}{d_{0}}$)^*n*^
Zhang et al.	Type I Portland cement W/B of 0.25~0.50	0	Performance evaluation methods. K-fold cross-validation	A MOLSSVR hyperparameter tuning system based on RBF core is developed.
Aliabdo et al.	Type I Portland cement W/B of 0.30	0, 50, 100	Permeability. Strength indices.	The relationship between permeability and strength performance is established.
Hatanaka et al.	Portland cement W/B of 0.22	100	Compressive strength. Porosity.	The compressive strength of recycled aggregate porous concrete can be roughly calculated from the porosity and the strength of broken concrete.
Rasiah et al.	Ordinary Portland cement and ground granulated blast furnace slag W/B of 0.33	100	Compressive strength. Porosity. Permeability.	* σ * = *m**σ_R·_*exp(*np*)
Lu et al.	Type I Portland cement W/B of 0.40	25, 50, 75, 100	Image analysis technique. Thermal conductivity. Compressive strength. Porosity. Permeability.	The use of waste glass or RA to replace the NA resulted in the decrease in the density and weak bonding between the paste and the aggregates.
Zhang et al.	Portland cement and fly ash W/B of 0.28	100	The crushing index of RA. Permeability. Strength indices.	The compressive strength, flexural strength and static elastic modulus of POC decrease significantly with the increase in crushing index of RA.
Naderpour and Mirrashid	—	—	The adaptive neuro-fuzzy inference system. Modeling of the ANFIS. Sensitivity analyses. The mathematical framework of the model.	An ANFIS model with six Gaussian membership functions for each input variable and six fuzzy rules
Chen et al.	Ordinary portland cement and fly ash W/B of 0.30	100	Splitting tensile strength. Compressive strength. Porosity. Permeability. BP neural network model.	Unilateral prediction model and Bilateral prediction model

Where *σ* is the compressive strength of POC (MPa), *σ*_0_ is the compressive strength when the porosity is 0 (MPa), *V* is the porosity (%), *b* is the empirical constant, *B* is the empirical constant, *m* and *n* are new material constants for porous concrete, *e* is the elasticity modulus (Pa), *φ* is the effective porosity (%), *d*_0_ is the smallest aggregate size used in the study, *d* is the average aggregate size used in the study, *σ*_R_ is the compressive strength of RPOC (MPa).

## 2. Experimental

### 2.1. Raw Materials

#### 2.1.1. Cement

The Ordinary Portland cement (P·O 42.5 in Chinese Standard, purchased from Shanshui Group Co., Ltd., Shandong, China) was used as a cement in this study. Its chemical composition (df-1000 X-ray fluorescence of Beijing jitian Co., Ltd., Beijing, China) and physical properties are shown in Table 2 and Table 3, respectively.

#### 2.1.2. Aggregates

Natural aggregate is granite gravel (acquired from the Samsung Company, Shandong, China), as shown in Figure 2a. Recycled aggregate is sorted from construction and demolition waste by sorting equipment (acquired from the lvfan building materials company, Shandong, China), as shown in Figure 2b. The composition of recycled aggregate is shown in Table 4, and the phase composition (D8 advance X-ray polycrystalline diffractometer of AXS Brooke Co., Ltd., Karlsruhe, Germany) is shown in Figure 3. According to the grade requirements of the Chinese standard GBT 25177, it belongs to Class III recycled aggregate (low-grade recycled aggregate). The particle size of the aggregates used is 5–20 mm, and the particle size distribution of aggregates with the mixture is shown in Figure 4. The summary of the physical and mechanical properties of the aggregates is shown in Table 5 (determined according to GB specification).

#### 2.1.3. Water Reducer Agent and Water

Polycarboxylate superplasticizer (PS, acquired from Subote Co., Ltd., Jiangsu, China) was used to improve the fluidity of fresh cement paste. The mixing water is tap water.

### 2.2. Mix Proportion and Preparation

The mix proportion of POC was determined by the unit volume method (aggregate volume + cement paste volume + design porosity = 1), and the parameters were selected as follows:Water–cement ratio: according to previous research results [38], the optimal water-cement ratio of POC is around 0.25;Replacement rate of RA: The total volume of aggregate was replaced by 0%, 25%, 50%, 75% and 100% of RA, respectively. The POC with NA was denoted as NPOC, and the POC with RA was denoted as RPOC-x (x = replacement rate of recycled aggregate, %);Design porosity: Design porosity: 15%, 20%, 25%, 30% were selected with reference to actual engineering requirements [39];Dosage of PS: PS was added to control the fluidity of the concrete mixture. According to the existing literature [40] and many tests, it is determined that when the fluidity of the cement paste is between 180 mm and 200 mm, POC is easy to form.

It should be noted that in order to avoid the influence of water–cement ratio due to the absorption of mixing water by aggregates, all aggregates need to be saturated with water before mixing. The mix proportion of POC is shown in Table 6.

In this study, raw materials were weighed according to the mix proportion and then added to the concrete single horizontal shaft forced concrete mixer (HJW-60L type, Chuancheng Environmental Protection Technology Co., Ltd., Shandong, China). Firstly, cement and aggregate were added and mixed for 0.5 min, then the mixture of water and water reducer was added and mixed for a further 1.5 min until the surface of the concrete mixture appears metallic luster. The fresh concrete mixture was poured into a mold of 150 × 150 × 150 mm^3^ for compressive strength and porosity test. It should be noted that the concrete mixture was loaded into the mold in three layers, and the tamping rod (diameter 16 mm) was inserted clockwise 25 times after each layer was filled. Finally, the particles protruding from the surface of the mold were removed, and the concave parts were filled with appropriate particles. At the same time, the cement paste (according to the mix proportion of POC) was prepared by cement paste mixer (NJ160 type, Cangzhou Huayang Testing Machine Manufacturing Co., Ltd., Cangzhou, China) and put into a prism mold of 40 mm × 40 mm × 160 mm. All molds were covered with preservative film and kept at room temperature for one day. After 24 h, the samples were taken out of the molds and transferred to the curing room with a temperature of 22 ± 2 °C and relative humidity greater than 90 R.H. until the required age. The specific process is shown in Figure 5.

## 3. Tests of Properties of Concrete

### 3.1. Compressive Strength

The compressive strength test of POC referred to “Compressive strength test of concrete” GB/T 50081, and the specimens of 150 × 150 × 150 mm^3^ were completed by a concrete compression testing machine (DYE-2000 type, Cangzhou Zerui Testing Instrument Co., Ltd., Hebei, China) at the age of 28 days.

The compressive strength test of cement paste referred to “strength test of cement mortar” GB/T 17671, and the specimens of 40 mm × 40 mm × 160 mm were completed by an automatic bending and compression testing machine (YAW-300c type, Jinan Hengxu Testing Machine Technology Co., Ltd., Shandong, China) at the age of 28 days.

### 3.2. Porosity

The porosity of POC was measured using the drainage method in accordance with the “Test method for porosity of porous concrete” report of the Japan environmental protection concrete research committee. Firstly, the concrete specimens were transferred from the curing room to the container, then the specimens were covered with water and soaked for 24 h, and the mass of the specimens in water was weighed by the electronic hook scale. After that, the specimens were heated in the oven at 105 °C for 24 h. The test process of porosity was shown in Figure 6, and the porosity could be calculated by Equation (1).
(1)$$P=1-\frac{\frac{W_{1}{-W}_{2}}{\rho_{0}}}{V}\;\times \;100\%$$
where, *P* was the porosity (%), *W*_1_ was the mass of the specimen dried to constant weight in the drying oven (kg), *W*_2_ was the mass of the specimen immersed in water (kg), *ρ*_0_ was the density of water (kg/m^3^) and *V* was the apparent volume (m^3^).

### 3.3. SEM Analysis

To further observe the interface transition zone (ITZ) between cement paste and aggregate, a scanning electron microscope (SEM, JEOL 7500F type, Tokyo, Japan) was used to characterize the microstructure of the specimen.

## 4. Results and Discussions

### 4.1. Influence of RA on Compressive Strength of POC

Table 7 shows the test results and standard deviations, and the compressive strength of POC with a different replacement rate of RA are shown in Figure 7. It is observed that the compressive strength of POC decreases in varying degrees after mixing RA. The specific changes are as follows: when the replacement rate of RA is less than 25%, the compressive strength of POC hardly changes (the strength fluctuates slightly due to test error and other factors), and then the compressive strength gradually decreases with the increase in replacement rate. The main reason for this phenomenon is that the low-grade RA contains waste concrete blocks, bricks and other impurities (as shown in Table 3), and the strength and grain shape are uneven, which has a negative impact on the strength of products. However, when the replacement rate is low, an appropriate amount of low-grade RA is not enough to affect the overall performance of POC. It proves that the POC can be mixed with less low-grade RA under the condition of constant engineering demand.

As shown in Figure 7, the compressive strength of various types of POC decreases with the increase in porosity. This obvious rule can be used to establish the prediction formula for compression strength of POC with low-grade RA. At present, it was confirmed that porosity is an important factor affecting the compressive strength of POC, and the relationship between the two can be accurately calculated by the compressive strength and porosity relationship of brittle, porous materials, as shown in Equation (2). The relevance of this formula is detailed in previous research [41].
*σ* = *σ*_0_exp (−*DP*)(2)
where *σ* was the compressive strength of POC (MPa), *σ*_0_ was the compressive strength when the porosity is 0 (MPa), *P* was the porosity (%), *D* was the empirical constant.

The above formula reflects the exponential relationship between the porosity and compressive strength of POC, where *σ*_0_ is referred to as the compressive strength of the material when the porosity is 0. However, for POC materials, it is difficult to achieve a seamless connection between different phases because it contains no or less fine aggregate in its composition. The SEM images of ITZ between aggregate and cement paste in different types of POC are shown in Figure 8. It is observed that there are a large number of micropores in the bond between the aggregate and the cement paste, and these inevitable pores make it difficult to measure the compressive strength of POC when the porosity is 0. In response to this phenomenon, Japanese scholar Hatanaka [42] believes that although it is impossible to prepare POC specimens with 0 porosity, an approximate relationship curve can be obtained by numerical substitution, and the strength substitution theory based on cement paste is proposed through experimental research. The results show that the strength of POC with 0 porosity can be approximately replaced by the strength of cement paste with the same volume when there is little difference between the strength of aggregate and cement paste. Therefore, in the past, when Equation (2) was used to calculate the compressive strength of POC, the strength of cement paste was often needed.

However, the types of RA are complex, some of which are comparable to natural aggregate, and some may contain glass, bricks and other debris [43]. When the grade of RA used is low, or the replacement rate is high, the difference between the strength of aggregate and cement paste in POC may be too large. In this case, if Equation (2) is still used to calculate the compressive strength of RPOC and the strength of cement paste is taken as *σ*_0_, a larger error will occur. It can be seen from Figure 9 that no matter what the aggregate type is, the calculation curve obtained by Equation (2) can always reflect the strength change trend of POC. However, with the increase in the replacement rate of recycled aggregate, the difference between the calculated curve and the actual compressive strength increases. This is closely related to the value of *σ*_0_, which proves that the Equation (2) of cement paste strength as *σ*_0_ does not take into account the variety of aggregate. Therefore, it can not be directly used in the preparation of POC with low-grade RA, and it needs to be modified properly.

### 4.2. Relationship between NPOC and RPOC

In the study of RA, it can be regarded as poor-quality NA [44]. Similarly, there is a certain relationship between RPOC and NPOC. The compressive strength of POC with a different replacement rate of RA at the same design porosity is shown in Figure 10. It can be seen that the linear regression coefficient R^2^ of each line is close to 1. The results show that when the target porosity is constant, there is a significant relationship between the compressive strength of POC with a different replacement rate of RA, and it can be expressed by a linear relationship. Therefore, the influence coefficient *β* of RA is introduced to establish the formula of compressive strength of RPOC, as follows:*σ_R_* = *β*·*σ*(3)
where *σ_R_* was the compressive strength of RPOC (MPa), *β* was the influence coefficient of RA.

*β* is the negative influence of RA on the compressive strength of POC, and its value is determined by the quality and replacement rate of RA. As shown in Figure 11, the influence coefficient *β* first increases and then decreases with the increase in the replacement rate of RA. In order to simplify the calculation, linear regression was selected for the replacement rate of RA and *β*. It can be seen that there is a good linear relationship between the points, and the influence coefficient *β* can be calculated by Equation (4). In the regression equation in Figure 11, the replacement rate of RA determines the change in the slope of the equation, and the intercept on the *Y*-axis represents other factors that affect *β*, such as the crush value of RA, water absorption, etc. Since the same RA is used in this study, the intercept of the regression equation remains unchanged. Based on the above research, the calculation formula for the compressive strength of RPOC is shown in Equation (5).
*β* = −0.005*α* + 1.0507(4)
*σ_R_* = (−0.005*α* + 1.0507)·*σ*(5)
where *α* was the replacement rate of RA (%).

As shown in Figure 12, the error between the actual strength and the strength calculated by Equation (5) is almost no more than 15%. It is verified that Equation (5) can be used to express the compressive strength of POC when the water–binder ratio is 0.25 and the quality of RA is similar to Class III in Chinese standard. In the same way, this method can also be used to calculate the RPOC prepared with other mix proportions.

### 4.3. Compression Damage Mechanism of POC

Although the compressive strength of POC with low-grade RA can be predicted by the above results, the calculation process is relatively complicated, and not all the cement paste strengths are measured. Therefore, the idealized method is used in this series to try to establish a simple and effective prediction model.

In order to more directly observe the morphological characteristics of POC when it is damaged under compression, the POC with 0% and 100% RA replacement rates are selected to compare the fracture surface under compression damage, as shown in Figure 13. It can be observed that the compression fracture surface of NPOC is relatively flat; there are both cement paste fractures and many crushed aggregate sections, which indicates that the structural damage of POC is mainly caused by the damage of aggregate and the aggregate plays a full supporting role. On the other hand, the compressive fracture surface of RPOC is uneven, and most of the aggregates are intact, and the failure mainly occurs in the cement paste between the aggregates. The reason for this phenomenon is that the surface of low-grade RA is wrapped with a layer of old cement mortar, which absorbs a lot of water during concrete mixing, and reduces the bond strength of cement paste. Under the action of pressure, the aggregate is easy to be peeled off, and the fracture surface is uneven.

In addition, due to their high crushing value, some bricks are the first to break under the impact of external force. Therefore, the mechanical properties of POC with low-grade RA are weak.

Based on this, the compression damage mechanism of POC is preliminarily studied, and the conceptual diagram of the whole process is shown in Figure 14. When subjected to pressure, all parts of the same plane in the POC are stressed at the same time, and the cement paste between the aggregates often breaks first due to the small bonding area. When the paste breaks, part of the pressure will be transferred to the lower plane, and the rest of the pressure will be borne by the aggregate of the layer. More importantly, the higher the strength of the cement paste, the longer the bearing pressure for the upper aggregate, such as NPOC. On the contrary, the strength of cement paste around the low-grade RA is low, and it will be destroyed without the support of the upper aggregate and continue to transfer the pressure downward, which eventually leads to the destruction of RPOC in a short time.

### 4.4. Relationship between Compressive Strength of RPOC and Bonding Area of Aggregate

Based on the research results of the compressive damage mechanism, it is found that the compressive performance of POC with low-grade RA is largely determined by the bonding area between aggregates. In order to explore this relationship, an ideal sphere model of RPOC is proposed. The low-grade recycled aggregates are assumed to be spheres of the same size, and the cement paste is also idealized to be evenly distributed on the surface of the spheres so as to calculate the bonding area between the aggregates. As shown in Figure 15, the radius of the ideal sphere is recorded as r, which is equal to 1/2 of the average particle size of RA, and the thickness of cement paste is t, which can be controlled in the design of mix proportion [45]. This modeling method is often used in ordinary concrete [46], but due to the existence of fine aggregate, the difference between the maximum aggregate size and the minimum aggregate size is large, and there is great randomness in spatial arrangement, so it is difficult to restore the real situation using the above modeling method. However, POC only contains coarse aggregate and has a large pore volume, so the randomness of aggregate arrangement is small. In order to ensure the same compression mechanism of RPOC on different surfaces, the ideal spheres are arranged into a cube, as shown in Figure 16a.

In this structure, the volume of aggregate contained in any cube unit is equal, which can represent the majority of POC to a certain extent. In addition, in order to make the cement paste adequately bonded, a certain pressure is often given to the concrete mixture during the preparation.

As shown in Figure 16b, the cement paste with two parts of spherical crowns is squeezed and settled under the action of external pressure. In this model, all the bonding is assumed to be in this state, and the squeezed paste does not participate in the discussion.

Since the compressive damage of POC with low-grade RA mostly occurs at the bonding area between aggregates, it is assumed that the fracture surface is the weakest part of the bonding area, as shown in Figure 17. It can be seen that the fracture area is equal to the bonding area between aggregates, which is defined as the effective fracture area. Therefore, the relationship between compressive strength and aggregate bonding area is established. In the numerical calculation, the effective fracture area of a single aggregate is equal to the bottom area of the squeezed part, which can be calculated by the spherical crown formula, as shown in Equations (6) and (7).
(6)$$A=\;\pi (t^{2}+2tr)$$
(7)$$t\;=\;\frac{V_{c}}{S_{c}}\;$$
where *A* was the effective fracture area (mm^2^), *r* was the radius of the ideal sphere of low-grade RA (mm), *t* was the thickness of cement paste (mm), *V_c_* was the volume of cement paste (mm^3^), *S_c_* was the surface area of the ideal sphere of low-grade RA (mm^2^).

As shown in Figure 18, the compressive strength of POC with low-grade RA increases with the increase in effective fracture area. The reason is that the increase in effective fracture area strengthens the bond between aggregates, or from another perspective, the pores in POC are filled by effective fracture area, which increases the overall compactness. It can be observed that there is a good exponential relationship between the compressive strength and the effective fracture area, which can be expressed by Equation (8). Coincidentally, this mathematical relationship is similar to the relationship between porosity and compressive strength of POC. It can be understood that Equation (8) is obtained by transforming the relationship between porosity and compressive strength to a certain extent, but it improves the deficiency that RPOC is difficult to obtain *σ*_0_. In order to verify the accuracy of the model, the calculated data are compared with the actual data in this paper and the other literature [33,34,47,48,49], as shown in Figure 19. The results show that the calculated values of the model are close to the actual values, especially when the concrete strength is low. It can be concluded that Equation (8) can predict the compressive strength of POC prepared with ordinary portland cement and low-grade RA in advance. However, because the influence of aggregate strength is not considered in the model, it is more suitable for the aggregate with similar quality to that used in this test, and there will be a large error when using recycled aggregate with different quality. Although the prediction model has some limitations, it verifies the relationship between effective fracture area and compressive strength and provides a new idea for solving the strength prediction problem of POC with low-grade RA.
*σ_R_* = 2.2712*e*^0.0099*A*^(8)

## 5. Conclusions

This study aims to provide a simple and effective prediction model for the compressive strength of POC with low-grade RA. The experiment starts with the existing compressive strength prediction model of POC, compares and analyzes its applicability to POC with low-grade RA, and improves the model by using the numerical fitting method. The compression fracture surfaces of NPOC and RPOC are selected to explore the compression damage mechanism, and based on this, a sphere model is used for physical modeling, and the compressive strength prediction formula of RPOC is obtained. The accuracy of the formula is verified by comparison with previous experimental data.

Based on the data obtained, the following conclusions can be drawn:The results show that the compressive strength of POC decreases with the increase in low-grade recycled aggregate. However, when the replacement rate is low, the change in strength is not obvious. On the contrary, the overall compactness will be increased due to the grain shape of the aggregate, and the strength will be improved to a certain extent;When the target porosity is constant, the compressive strength of POC with different replacement rates of low-grade RA can be expressed by a linear relationship. The influence coefficient of RA proposed in this paper based on the replacement rate of RA can be effectively used to express the compressive strength relationship between NPOC and RPOC. This solution can also be used for POC prepared from different grades of RA;The results on the compression damage mechanism of POC with low-grade RA show that the cement paste fractures first when the RPOC is compressed. Because the cement paste on the surface of low-grade RA is weakly bonded, most of the aggregate is peeled off without playing a supporting role, so it is easy to be damaged under pressure. This result can be understood as determining the compressive strength of POC is not the average grade of raw materials, but the lowest grade;The compressive strength of RPOC is related to the bonding paste between aggregates. When the paste material remains unchanged, the compressive strength of RPOC increases with the increase in bonding area, and the two can be expressed by an exponential relationship;The compressive strength of POC with low-grade RA can be accurately calculated by the compressive strength prediction formula based on the ideal sphere model.The shortcoming of the proposed structure analysis and modeling method of RPOC is that the diversity of recycled aggregate is not considered. Low-grade RA has different particle shapes, and the real scene cannot be reflected by the same physical model. Therefore, in further study, the authors will focus on the development of expression models of different RA, such as the method of polar coordinate axes, in order to more truly reflect the bonding state between aggregates in RPOC.

## Figures and Tables

**Figure 1 materials-14-03871-f001:**
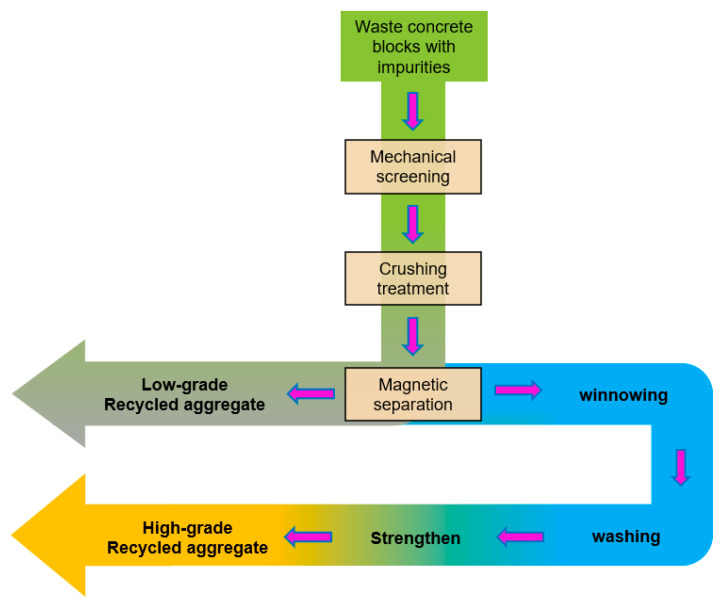
Common production process of recycled aggregate.

**Figure 2 materials-14-03871-f002:**
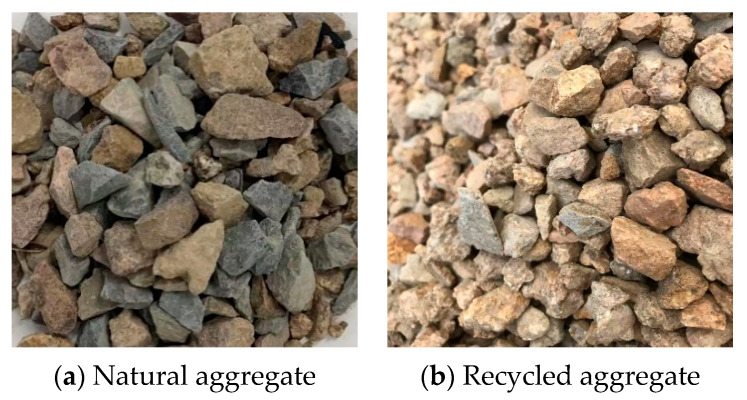
Appearance of natural aggregate and recycled aggregate.

**Figure 3 materials-14-03871-f003:**
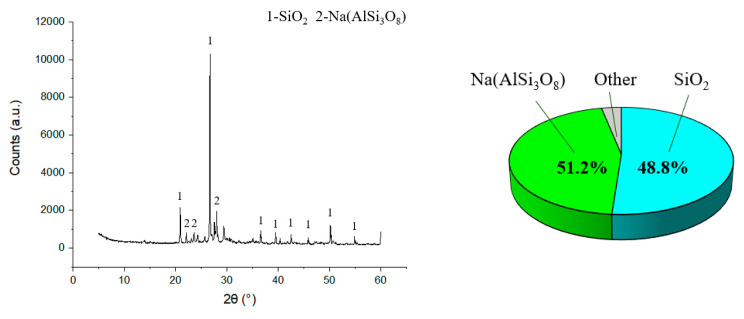
Phase composition of recycled aggregate.

**Figure 4 materials-14-03871-f004:**
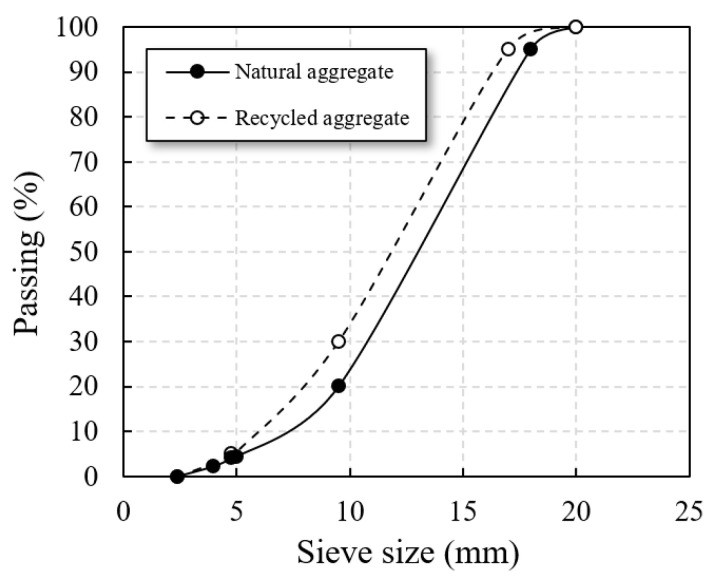
Particle size distribution of aggregates with mixture.

**Figure 5 materials-14-03871-f005:**
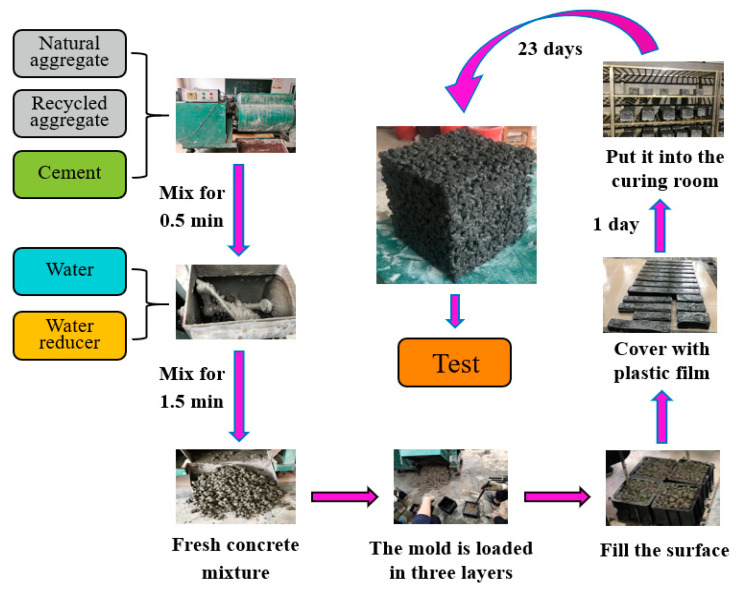
Flow chart of porous concrete preparation.

**Figure 6 materials-14-03871-f006:**
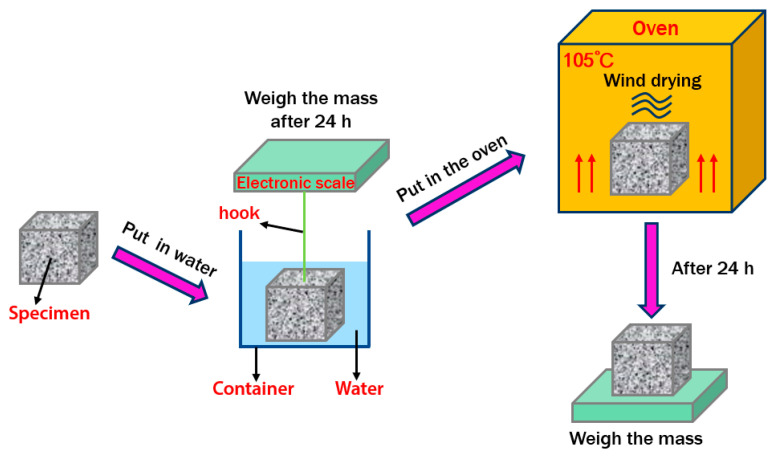
Flow chart of porosity test for porous concrete.

**Figure 7 materials-14-03871-f007:**
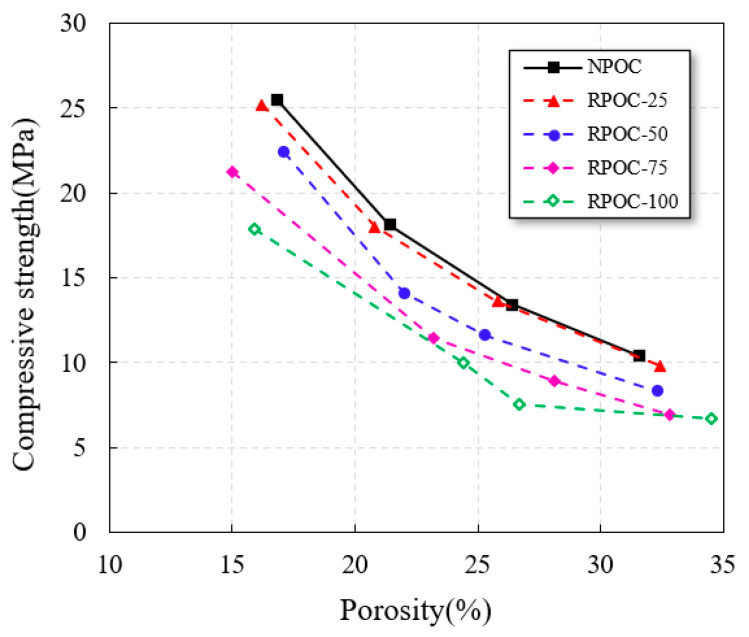
The compressive strength of porous concrete changes with the replacement rate of recycled aggregate.

**Figure 8 materials-14-03871-f008:**
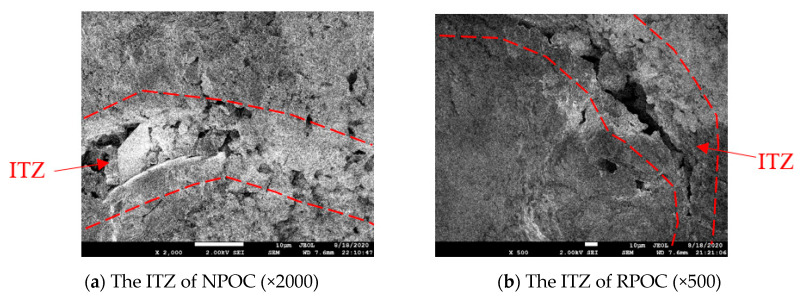
SEM images of ITZ between aggregate and cement paste in different types of porous concrete.

**Figure 9 materials-14-03871-f009:**
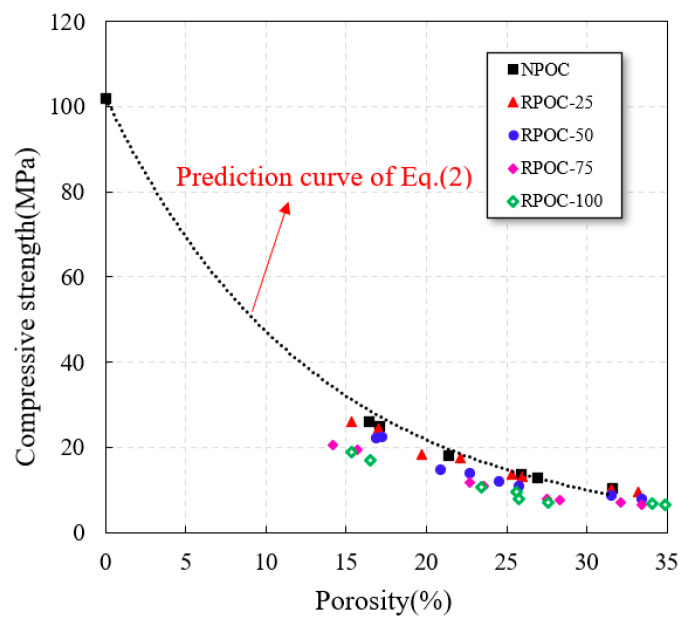
Relationship between actual compressive strength of various types of porous concrete and calculation curve of Equation (4).

**Figure 10 materials-14-03871-f010:**
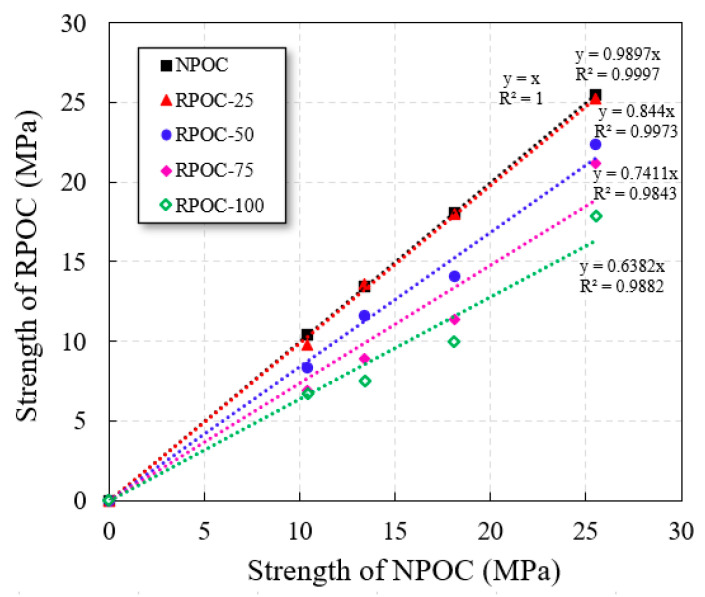
Relationship between compressive strength of porous concrete with a different replacement rate of RA.

**Figure 11 materials-14-03871-f011:**
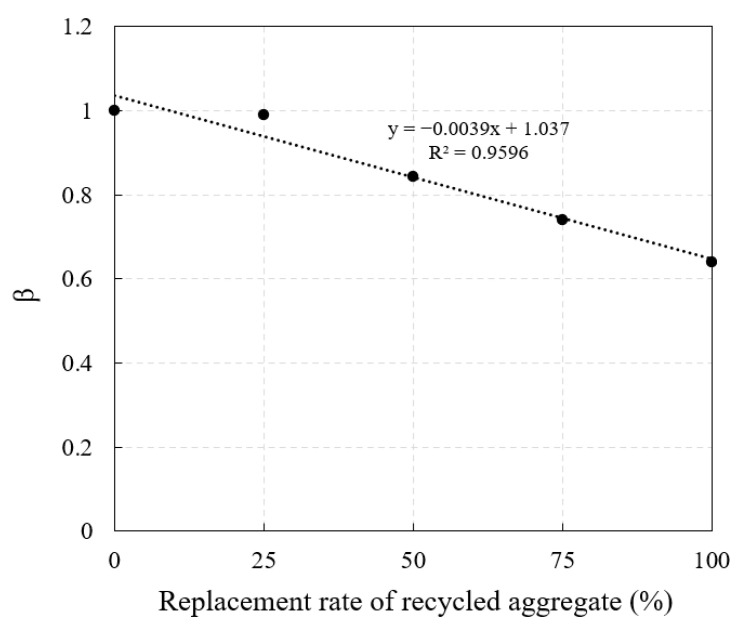
Relationship between influence coefficient of RA and replacement rate of RA.

**Figure 12 materials-14-03871-f012:**
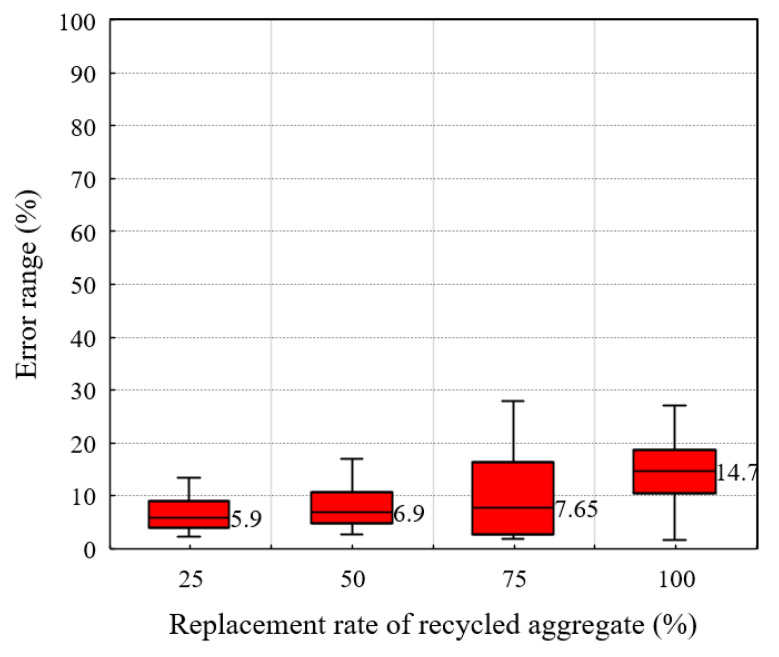
Error range of calculated strength.

**Figure 13 materials-14-03871-f013:**
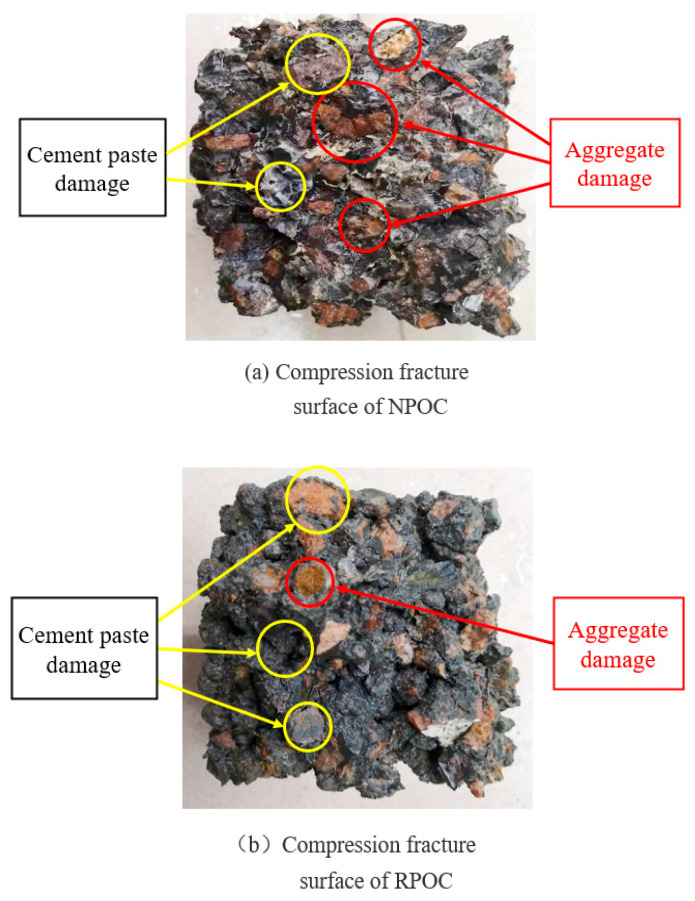
Compression fracture surface of porous concrete.

**Figure 14 materials-14-03871-f014:**
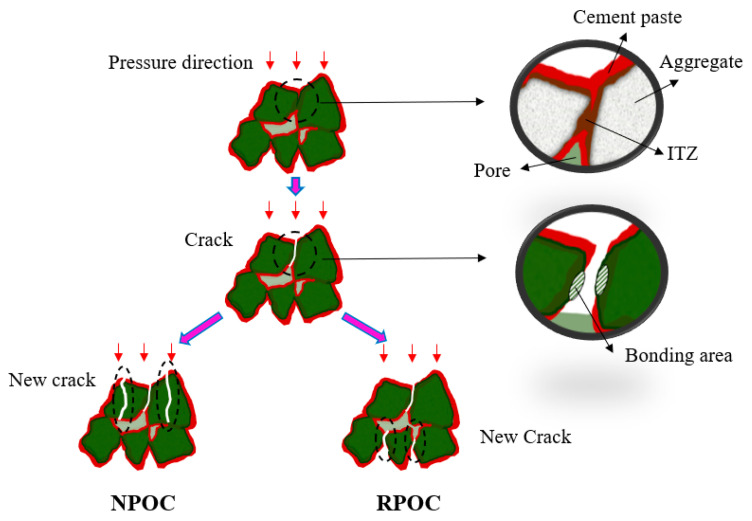
Conceptual diagram of porous concrete compression damage.

**Figure 15 materials-14-03871-f015:**
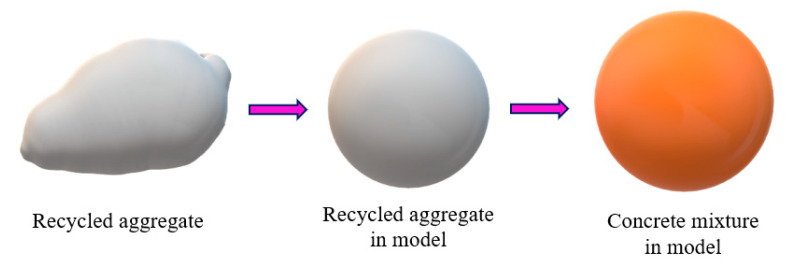
Ideal sphere model of low-grade RA.

**Figure 16 materials-14-03871-f016:**
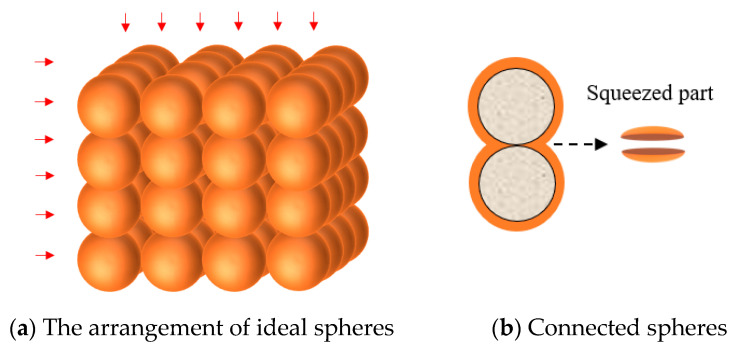
Structural model of RPOC.

**Figure 17 materials-14-03871-f017:**
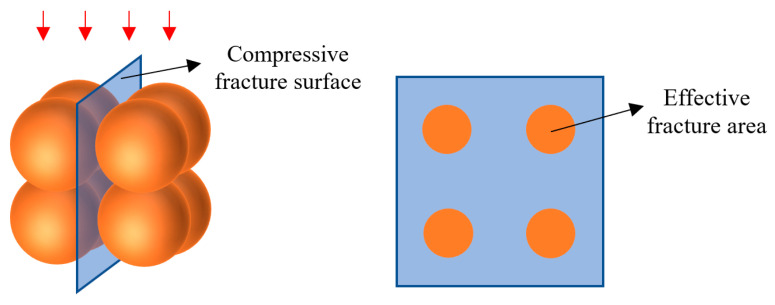
Fracture surface of compression damage.

**Figure 18 materials-14-03871-f018:**
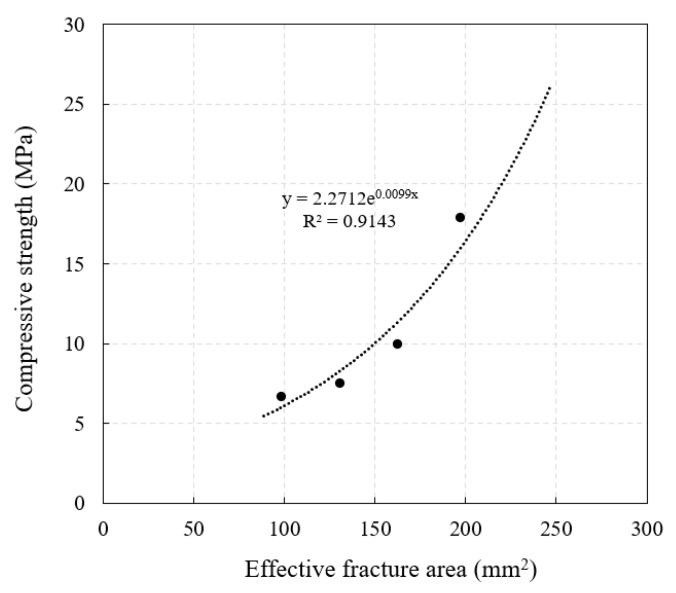
Relationship between compressive strength and effective fracture area.

**Figure 19 materials-14-03871-f019:**
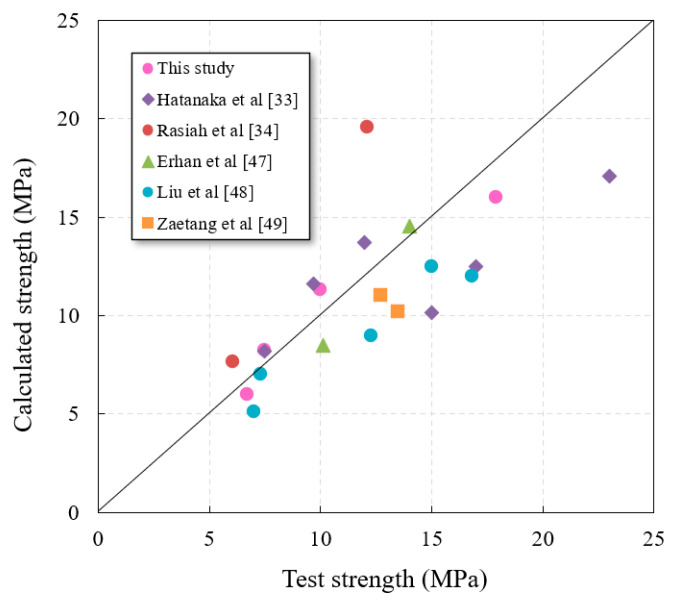
Relationship between calculated strength and actual strength.

**Table 2 materials-14-03871-t002:** Chemical composition of cement (%).

SiO_2_	AI_2_O_3_	Fe_2_O_3_	CaO	MgO	SO_3_	Na_2_O	LOSS
21.51	5.98	3.94	60.72	1.30	2.68	2.84	1.03

**Table 3 materials-14-03871-t003:** Physical and mechanical properties of cement.

Density (g/cm^3^)	Specific Surface Area (m^2^/kg)	Setting Time (min)		Flexural Strength (MPa)		Compressive Strength (MPa)	
		Start	Final	3 d	28 d	3 d	28 d
3.13 ± 0.02	342 ± 5	182 ± 2	251 ± 2	4.7 ± 0.5	7.5 ± 0.5	21.8 ± 1	47.6 ± 1

**Table 4 materials-14-03871-t004:** Composition of recycled aggregate.

Materia	Constituents (% by Weight)			
	Old Concrete	Natural Stones	Clay Bricks	Other Impurities (Glass, Wood, Pitch, Plastic, Paper, etc.)
Recycled aggregate	81.8	9.1	7.7	1.4

**Table 5 materials-14-03871-t005:** Physical and mechanical properties of aggregates.

Type of Aggregates	Physical and Mechanical Properties					
	Aggregate Size (mm)	Bulk Density (kg/m^3^)	Oven-Dried Particle Density (kg/m^3^)	Water Absorption (%)	Void Ratio (%)	Crushing Value (%)
NA	5~20	1327	2632	1.18	46.2	9.7
RA	5~20	1288	2421	3.20	45.5	26.1

**Table 6 materials-14-03871-t006:** Mix proportion of porous concrete.

Type	W/C	Design Porosity (%)	Natural Aggregate (kg/m^3^)	Recycled Aggregate (kg/m^3^)	Cement (kg/m^3^)	Water (kg/m^3^)	PS ^a^ (%)
NPOC	0.25	15	1426	0	627	157	0.30
		20	1426	0	539	135	0.30
		25	1426	0	451	113	0.30
		30	1426	0	363	91	0.30
RPOC-25	0.25	15	1063	354	608	152	0.32
		20	1063	354	520	130	0.32
		25	1063	354	432	108	0.32
		30	1063	354	344	86	0.32
RPOC-50	0.25	15	705	705	590	148	0.35
		20	705	705	502	126	0.35
		25	705	705	414	104	0.35
		30	705	705	327	82	0.35
RPOC-75	0.25	15	350	1050	573	143	0.37
		20	350	1050	485	121	0.37
		25	350	1050	397	99	0.37
		30	350	1050	309	77	0.37
RPOC-100	0.25	15	0	1391	555	139	0.40
		20	0	1391	467	117	0.40
		25	0	1391	379	95	0.40
		30	0	1391	291	73	0.40

^a^ The dosage of PS is the percentage of cement.

**Table 7 materials-14-03871-t007:** Test result.

Type	Design Porosity (%)	Actual Porosity (%)	Standard Deviation (%)	Compressive Strength (MPa)	Standard Deviation (MPa)
NPOC	15	16.8	1.12	25.5	0.93
	20	21.4	1.26	18.1	0.87
	25	26.4	0.89	13.4	1.13
	30	31.6	1.58	10.4	0.69
RPOC-25	15	16.2	1.26	25.2	0.77
	20	20.8	0.63	18.0	1.05
	25	25.8	0.94	13.6	1.14
	30	32.4	1.48	9.8	0.89
RPOC-50	15	17.1	0.83	22.4	1.22
	20	22.0	1.22	14.1	0.55
	25	25.3	1.18	11.6	0.71
	30	32.3	1.41	8.3	0.89
RPOC-75	15	15.0	0.77	21.2	1.14
	20	23.2	0.55	11.4	0.95
	25	28.1	1.34	7.9	0.84
	30	32.8	1.34	6.9	0.89
RPOC-100	15	15.9	0.89	17.9	1.10
	20	24.4	0.71	10.0	0.71
	25	26.7	1.18	7.5	0.77
	30	34.5	1.55	6.7	0.89

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
