# Peer review of "Prediction Model for Compressive Strength of Porous Concrete with Low-Grade Recycled Aggregate"

_materials, 2021, doi:10.3390/ma14143871_

Round 1

Reviewer 1 Report

The originality and the scientific value of the subject research can be better. 

Research area is Prediction model for compressive strength of porous concrete with low-grade recycled aggregate.

The manuscript has a total of 21 pages.

The experimental program is interesting, but some information is known. 

A number of research tasks and articles are devoted to the solved area. 
The article must significantly present new knowledge and recommendations for further research.
A possible revision of the document should be prepared in the MDPI journal template.

One of the weaknesses of the manuscript is that it does not solve the problem in a sufficiently complex way and focuses only on selected properties. 
It would also be appropriate to use, for example, sophisticated behavioural modelling methods, etc.

Concrete is a quasi-brittle material. It is important to study mechanical parameters comprehensively. Mechanical parameters include, in particular, compressive strength, tensile strength, modulus of elasticity or fracture energy. 

I recommend improving the introduction section. There is extensive research in the solved area, which needs to be better explained within the solved research (aggregate, mechanical properties, numerical analysis, ANN) of properties of concrete.  In particular, readily available references should be mentioned. Among the interesting also reviews and articles from the recent time and solved areas are:

Moreno-Juez, J.; Tavares, L.M.; Artoni, R.; Carvalho, R.M.d.; da Cunha, E.R.; Cazacliu, B. Simulation of the Attrition of Recycled Concrete Aggregates during Concrete Mixing. Materials 2021, 14, 3007.

Sucharda, O. Identification of Fracture Mechanic Properties of Concrete and Analysis of Shear Capacity of Reinforced Concrete Beams without Transverse Reinforcement. Materials 2020, 13, 2788. 

References are not in MDPI template format.
Figure 1 must be enlarged.
Table 2 - also add the standard deviation or coefficient of variation.
Figure 5 must be enlarged.

Equations 1 and 2 are not necessary - they are basic.
The Discussion chapter must be presented separately.
It is necessary to add standard deviations to Fig.7 in a suitable way, eg a new table
Fib.8 add a zoom or scale to the figure caption
Fig.13 it is necessary to enlarge the picture.

The achieved results must be discussed in the context of the current state of affairs. For now, the manuscript has limited informative value.

What's new for further research?

Overall, it is necessary to improve the presentation of the results of the research and increase the informative value of the results.
Expand the discussion and conclusion. Research results must be clearly presented in the context of current results.

The manuscript must be revised.

Reviewer 2 Report

Review:

materials-1260733

The aim of this paper is of high interest. It is clearly presented. I consider that the subject of the manuscript is interesting.

  1. Introduction:Authors should include the updates and differences of their work with respect to others on the same topic.
  2. Results and discussion:   No statistical data (CV, SD) are given in figures and tables and authors should include the number of samples used.
  3. Conclusions: Authors should improve the explain on the conclusions.

 I recommend accept after minor revision

Reviewer 3 Report

The revised manuscript "materials-1260733" is describing very interesting problem of using recycled aggregates as a aggregates in porous cementitious mixtures. The article has scientific soundness and it is interesting for the readers. However it might be beneficial to make some improvements, such:

Names of the Authors which works are reffer should be italics.
Remark for the Introduction: maybe the previous works should be presented in table with column labels 1. Authors 2. Type of cementitious materia 3. Volume of RA 4. tests performed 5. Main findings. It will be more clear for the reader. And based on this it will be easier to understand the novelty of the research.
Figure 2 - the scale in the figure will be beneficial for someone to see how big this aggregates are. 
Reviewer is curious aobut figure 4. Because if we use recycled aggregate it is sometimes mixture of a few aggregates and cement matrix. Thus how the authors achieved the pure aggregate instead of aggregates with "admixtures" (sometimes it might be elements)
Lane 223-230 The reviewer would like to know why the Authors have not done the test (firstly) bending test and then (secondly) the compression test of two halves of the specimens. It is more like a question instead of the remark. 
Lane 257 The authors have not present before how such RA look like. Thus this sentence is a little bit hard to understand without previous explanation.
Figure 10. The reviewer is curious how R^2 were calculated because for example for the pink function with the equation y=0.7411x R^2 is evaluated as 0.9843 which is very high and all the measured points should lie on the line but we can se some points which has about 18 MPa streght of NAEC and 12 MPa strenght of RACE but according to the function equation it should have more than 13.

The main issue with this article is lack of deeper investigation of the aggregates used. Recycled Aggregates are very often characterized that in case of waste concrete they have also old ITZ "inside" between the aggregate and cement matrix from the old concrete. Also problematic is using a few different waste materials in this case. RA are were different even if they are from the same material. Using plenty of different materials may affect the results and increase an error of calculation, Thus in analyzes of such properties it is better to decrease the number of variables. 

The article is valuable for the academic sociaty thus it can be published after revising it. 

Round 2

Reviewer 1 Report

The changes made the improvement of the manuscript.

The research area and results are from the context of the manuscript can better understand.

The results of the research and information value of the manuscript can be evaluated overall well.

The manuscript can be published in the journal.

Reviewer 3 Report

In Reviewer's opinion the article benefits from the revision. Thus it can be published.